# Harnessing the Cross-Neutralisation Potential of Existing Antivenoms for Mitigating the Outcomes of Snakebite in Sub-Saharan Africa

**DOI:** 10.3390/ijms25084213

**Published:** 2024-04-11

**Authors:** Suyog Khochare, Anurag Jaglan, U. Rashmi, Paulomi Dam, Kartik Sunagar

**Affiliations:** Evolutionary Venomics Lab, Centre for Ecological Sciences, Indian Institute of Science, Bangalore 560012, Karnataka, India; suyogk@iisc.ac.in (S.K.); anuragjaglan@iisc.ac.in (A.J.); rashmi1@iisc.ac.in (U.R.); paulomidam@iisc.ac.in (P.D.)

**Keywords:** snake venoms, antivenom, in vitro binding, median lethal dose, cross-neutralisation potency

## Abstract

Over 32,000 individuals succumb to snake envenoming in sub-Saharan Africa (sSA) annually. This results from several factors, including a lack of antivenom products capable of neutralising the venoms of diverse snake species in this region. Most manufacturers produce polyvalent antivenoms targeting 3 to 16 clinically important snake species in sSA. However, specific products are unavailable for many others, especially those with a restricted geographic distribution. While next-generation antivenoms, comprising a cocktail of broadly neutralising antibodies, may offer an effective solution to this problem, given the need for their clinical validation, recombinant antivenoms are far from being available to snakebite victims. One of the strategies that could immediately address this issue involves harnessing the cross-neutralisation potential of existing products. Therefore, we assessed the neutralisation potency of PANAF-Premium antivenom towards the venoms of 14 medically important snakes from 13 countries across sSA for which specific antivenom products are unavailable. Preclinical assays in a murine model of snake envenoming revealed that the venoms of most snake species under investigation were effectively neutralised by this antivenom. Thus, this finding highlights the potential use of PANAF-Premium antivenom in treating bites from diverse snakes across sSA and the utility of harnessing the cross-neutralisation potential of antivenoms.

## 1. Introduction

In addition to being a land of many breathtaking landscapes, ranging from equatorial rainforests and savannas to tropical deserts, Africa is home to diverse fauna, including several iconic species of mammals and reptiles. Similarly to the rest of the world, habitat destruction and ever-increasing anthropogenic encroachment in Africa are paving the way for numerous man–wildlife conflict scenarios. Snake–human conflict represents the epitome of the aforementioned scenario, as over 81,000 to 138,000 lives are lost to snakebite annually on a global scale [1]. Sub-Saharan Africa (sSA) alone contributes to one-fifth of the global snakebite statistics, with ~32,000 deaths annually [2,3]. Currently, the only available therapy for snakebite is the conventional antivenom produced via the hyperimmunisation of equines with the target snake venoms [4,5,6,7]. Although a handful of antivenom products are currently marketed to treat snake envenoming in sSA, they only target a few medically relevant species [8]. This leaves out numerous other snakes capable of inflicting life-threatening envenoming. Since antivenoms are unavailable to treat bites from these species, clinicians are left with no other option but to use non-specific antivenoms [7]. ‘Next generation’ snakebite therapy, which includes the recombinant production of broadly neutralising monoclonal antibodies, has several advantages over conventional antivenoms [9,10,11,12,13]. Even though recombinant antivenoms offer a promising solution for addressing the aforementioned issue, they require clinical validation, and hence, they are unlikely to be commercially marketed over the next decade. Therefore, there is a pressing need to innovate alternative strategies for treating bites from diverse snake species that can bear an immediate impact. Alternative strategies for snakebite treatment include the use of medicinal plant extracts [14], aptamers [10], peptides [15] and small-molecule inhibitors [16]. In addition, harnessing the cross-neutralisation potential of existing antivenom products could offer an immediate solution to this problem.

To address this shortcoming, we evaluated the cross-neutralisation potential of PANAF-Premium, an antivenom manufactured by Premium Serums and Vaccines Pvt. Ltd. (Pune, Maharashtra, India) for treating bites from 14 snake species (*Bitis arietans*, *B. gabonica*, *B. nasicornis*, *B. rhinoceros*, *Echis leucogaster*, *E. ocellatus*, *E. carinatus*, *Naja haje*, *N. melanoleuca*, *N. nigricollis*, *Dendroaspis polylepis*, *D. viridis*, *D. jamesoni* and *D. angusticeps*) in sSA. We used a murine model of snake envenoming to evaluate whether this antivenom is capable of offering cross-neutralisation against other clinically relevant snake species from 13 countries across sSA (viz., Botswana, Benin, Cameroon, Congo, Egypt, Limpopo, Nigeria, Northern Cape, RSA, Tanzania, Togo, Western Cape and Zimbabwe). In particular, we tested whether the antivenom is able to counter the toxicity inflicted by *N. anchietae*, *N. annulifera*, *N. ashei*, *N. katiensis*, *N. mossambica*, *N. nigricincta*, *N. nivea*, *N. nubiae*, *N. pallida*, *N. senegalensis*, *N. woodi*, *B. caudalis*, *E coloratus* and *E. pyramidum*. Therefore, this study aimed to evaluate the cross-neutralisation potential of Premium Serums’ PANAF-Premium antivenom against medically relevant snakes across sSA. The outcomes of this study can potentially guide the antivenom marketing and deployment strategies in sSA.

## 2. Results

### 2.1. Protein Concentration

The Bradford method was used to estimate the protein concentration of venom samples (Table 1).

### 2.2. Sodium Dodecyl Sulphate–Polyacrylamide Gel Electrophoresis (SDS-PAGE) Profiles of African Snake Venoms

SDS-PAGE analyses revealed significant interspecific and intraspecific differences in the venom profiles of African elapid and viperid snakes. While the elapid (genera *Naja* and *Dendroaspis*) venoms were predominantly dominated by low-molecular-weight toxins, bands corresponding to ~25, ~50 and ~75 kDa showed varying intensities across species (Figure 1). The venoms of African *Echis* spp. exhibited bright bands corresponding to high- (50–75 kDa), mid- (15–25 kDa) and low-molecular-weight (8–15 kDa) toxins (Figure 1). The venoms of both *Bitis* snakes were found to consist of large amounts of low-molecular-weight (8–15 kDa) toxins, while bands corresponding to mid- (~25–37 kDa) and high-molecular-weight (~75 kDa) ranges starkly varied between the two species.

### 2.3. The In Vitro Binding Potential of PANAF-Premium Antivenom

Results of indirect ELISAs for various African snake venoms suggested that PANAF-Premium antivenom exhibits a varying recognition potential towards them (Figure 2 and Figure 3). For the members of the Elapidae family, greater binding was observed towards the venoms of *N. nigricollis*, *N. ashei*, *N. woodi*, *N. nivea*, *N. annulifera*, *N. mossambica*, *N. nubiae*, *N. pallida*, *N. nigricincta* and *N. katiensis* (endpoint titre: 1:2500), whereas relatively lower recognition was noted towards the venoms of *N. anchietae* and *N. senegalensis* (titre of 1:500). Surprisingly, although the venom of *D. polylepis* is a part of the immunisation mixture, the binding of PANAF-Premium antivenom towards this venom was relatively poor (titre 1:500). The titre value is defined as the highest dilution of an antivenom at which the OD value is significantly higher than the negative control (mean OD of naive IgG + two times standard deviation).

In the case of vipers, PANAF-Premium exhibited better cross-reactivity towards the venoms of *E. coloratus* and *E. pyramidum* (endpoint titre: 1:2500) but relatively lower binding towards the *E. ocellatus* venom, which is a part of the immunisation mixture. PANAF-Premium exhibited a relatively poor venom recognition potential against the *B. arietans* venom (titre: 1:2500), which is a part of the immunisation mixture. As for the other member of this genus, *B. caudalis*, the antivenom did exhibit binding but with a relatively lower titre of 1:500 (Figure 3).

### 2.4. In Vivo Toxicity Evaluation of Snake Venoms from sSA

The potency of African snake venoms was assessed by intravenously injecting venom dilutions in CD-1 mice and observing mortality over a period of 24 h. Amongst all the elapid snakes in this study, the venom of *D. polylepis* was found to be most potent towards the mice, having an LD_50_ of 0.349 mg/kg (Figure 4, Appendix A). The median lethal dose of the rest of the African elapids of the genus *Naja* ranged between 0.480 and 2.55 mg/kg, wherein *N. nigricollis* was the most toxic cobra species, followed by *N. pallida* (0.538 mg/kg) and *N. woodi* (0.545 mg/kg). In contrast, the *N. anchietae* venom was the least toxic member of the genus *Naja* (LD_50_: 2.55 mg/kg) venoms and was characterised by a potency of 5.3× lower than its congener, *N. nigricollis* (LD_50_: 0.480 mg/kg; Figure 4, Appendix A).

Similarly, the toxicity profiles of various African viperid snake venoms were also estimated, which revealed higher potencies of *E. coloratus* and *B. caudalis* venoms, with an LD_50_ value corresponding to 0.34 mg/kg, followed by *B. arietans* (LD_50_: 0.66 mg/kg) and *E. ocellatus* (LD_50_: 0.70 mg/kg; Figure 5, Appendix A) venoms. Among all of the tested viperid venoms, the venom of *E. pyramidum* was found to be the least toxic to mice, with LD_50_ of 0.98 mg/kg (Figure 5, Appendix A).

### 2.5. In Vivo Neutralisation Potential of PANAF-Premium Antivenom

PANAF-Premium antivenom displayed a high neutralisation potential against the snake venoms included in the immunisation mixture. Against *N. nigricollis* and *D. polylepis* venoms, per millilitre of this antivenom product was capable of neutralising 33 LD_50_s (0.318 mg/mL) and 28 LD_50_s (0.208 mg/mL), respectively, in the mouse model of snake envenoming (Figure 6, Appendix A). PANAF-Premium antivenom also displayed considerable cross-reactivity towards the venoms of *Naja* spp. from sSA. The neutralisation efficacy of this antivenom was calculated to be 111 LD_50_s/mL (0.370 mg/mL) towards the venom of *N. woodi* when tested with a 3× LD_50_ challenge dose. This was the highest cross-neutralisation potential observed in this study. This was followed by the cross-reactivity towards the venom of *N. nubiae*, which was half as much as that towards the venom of *N. woodi* (49.58 LD_50_s/mL; 1.216 mg/mL). With the exception of the venoms of *N. mossambica* (16.57 LD_50_s/mL; 0.441 mg/mL), *N. katiensis* (11.16 LD_50_s; 0.331 mg/mL) and *N. annulifera*, the venoms of all other snakes were effectively cross-neutralised by PANAF-Premium antivenom above their claimed neutralisation potency (≥20 LD_50_s/mL).

Similarly, the venoms of all but one viperid snake (*B. caudalis*) from sSA were effectively neutralised by PANAF-Premium antivenom (Figure 7, Appendix A). The highest neutralisation efficacy was shown towards the venoms of *E. pyramidum* (69.44 LD_50_s/mL; 1.739 mg/mL) and *B. arietans* (66.69 LD_50_s; 0.878 mg/mL), of which the latter is a part of the immunisation mixture. This antivenom also exhibited cross-reactivity towards the venom of the other two members of the genus *Echis*, having a neutralisation potency of 49.58 LD_50_s/mL for both *E. coloratus* (0.340 mg/mL) and *E. ocellatus* (0.693 mg/mL) venoms, which was 2× greater than the marketed neutralisation potency of this antivenom (≥25 LD_50_s/mL).

## 3. Discussion

### 3.1. Variations in Toxicity Profiles of African Snake Venoms

Africa is home to numerous species of medically important snakes, primarily belonging to the Elapidae (genera *Naja* and *Dendroaspis*), Viperidae (genera *Bitis* and *Echis*), Colubridae (genus *Dispholidus*) and Pseudaspididae (genus *Pseudaspis*) families [17,18,19]. In this study, we assessed the toxicity profiles of the venoms of 13 elapids and 5 viperids, which are considered medically significant in various regions of sSA. Although we documented species-specific differences in the SDS-PAGE venom profiles of elapids under investigation, all of them were dominated by low-molecular-weight toxins (~5–15 kDa). Despite this, the toxicity of elapid venoms varied greatly when tested in the mouse model of snake envenoming. In vivo experiments revealed the highest toxicity for the black mamba (*D. polylepis*) venom amongst elapids, with an LD_50_ of 0.349 mg/kg. The venoms of *N. woodi*, *N. pallida* and *N. senegalensis* were also found to be quite potent against mice, with an LD_50_ ranging between 0.54 to 0.56 mg/kg. However, the toxicity of the *N. anchietae* venom was the lowest amongst *Naja* spp. (2.55 mg/kg). In the case of viperids, represented by the genera *Echis* and *Bitis* in this study, mid-molecular-weight and high-molecular-weight toxins constituted the majority of their venoms. The toxicity of the venoms of two of the viperids included in this study, namely *E. coloratus* and *B. caudalis*, was at par with that of the venom of black mamba, having a median lethal dose of 0.34 mg/kg when tested in mice. Considerable variation in venom toxicities between snake species is widely documented and can result in inefficient neutralisation of bites using conventional antivenoms [9].

### 3.2. Cross-Neutralisation Potential of PANAF-Premium Antivenom

One of the major challenges in snakebite mitigation in sSA is ensuring a continued supply of high-quality antivenoms. The SAIMR polyvalent antivenom, manufactured by South African Vaccine Producers (SAVP), is considered the ‘gold standard’ for the treatment of snakebites in sSA. However, owing to the exorbitant price of this antivenom, victims are forced to rely on alternative and affordable non-specific antivenom products, which often leads to serum sickness and fatal anaphylaxis. As a result of this conundrum, several studies have highlighted the use of the polyvalent antivenom manufactured against the ‘big four’ snakes of India (*N. naja*, *Bungaurs caeruleus*, *E. carinatus* and *Daboia russelii*) for treating snakebites in sSA [7,8]. Not helping the situation, another manufacturer, Sanofi Pasteur, halted the production of its antivenom product in 2014 due to low profit margins, further contributing to the observed supply and demand gap in sSA [7,20]. In addition, a few antivenom products manufactured and marketed at relatively lower prices by companies based in India are also available for snakebite treatment in sSA. One of them is PANAF-Premium antivenom, which is manufactured by Premium Serums and Vaccines Pvt. Ltd. (Pune, Maharashtra, India) against the venoms of as many as 14 medically important snakes in sSA (Section 4.1). Yet, this leaves out many other medically relevant species for which specific antivenom products do not exist. Next-generation antivenoms, which include recombinantly expressed antibodies targeting the evolutionarily conserved regions in major toxins, are capable of neutralising the venoms of distantly related snake species [12,13]. However, since they require clinical validation in humans, they are unlikely to be commercialised in the next 10 years. In contrast, harnessing the cross-neutralisation potential of currently available products against closely related species could be one of the immediate solutions to address the snakebite problem.

Hence, in this study, we estimated the in vitro binding and in vivo neutralising potency of PANAF-Premium antivenom towards the non-targeted snake species from 13 countries in sSA (Table 1). Our findings suggest that PANAF-Premium exhibits a significant in vitro recognition towards the tested elapid and viperid venoms (Figure 2 and Figure 3). The titre value of the venoms of most of the elapids, except *N. anchietae*, *N. senegalensis* and *D. polylepis* (1:500), was around 1:2500, suggesting increased binding. Similarly, with the exception of *B. caudalis*, whose venom is a part of the immunisation mixture (1:500), PANAF-Premium antivenom exhibited a titre value of 1:2500 against the venoms of viperid snakes (*E. coloratus*, *E. ocellatus*, *E. pyramidum* and *B. arietans*). When the PANAF-Premium product was tested for its efficacy in neutralising the venoms of medically important snakes in sSA (Table 1), we found that it exhibits considerable neutralisation efficacy towards the venoms of the majority of elapid snakes. The neutralisation potency of this product exceeds the marketed neutralisation potential of 20× LD_50_ for the snake venoms included in the immunisation mixture (*N. nigricollis* and *D. polylepis*), as well as against the venoms of other non-target snake species: *N. ashei*, *N. anchietae*, *N. nigricincta*, *N. nivea*, *N. nubiae*, *N. senegalensis*, *N. pallida* and *N. woodi*. Among all elapids tested in this study, the venoms of three species, namely *N. annulifera*, *N. katiensis* and *N. mossambica*, were not effectively neutralised by PANAF-Premium antivenom (neutralisation potency ranged between 11 and 16 LD_50_s or 0.3 and 0.4 mg/mL; Figure 6). In the case of the viperids under investigation, PANAF-Premium antivenom was able to effectively neutralise the venoms of the representatives of the genus *Echis* tested in this study (*E. ocellatus*, *E. coloratus* and *E. pyramidum*) and one of the representatives of the genus *Bitis* (*B arietans*), with a neutralisation potency ranging between 50 and 70 LD_50s_ or 0.3 and 1.7 mg/mL (Figure 7, Appendix A). However, it completely failed to protect mice from the toxic effects of the *B. caudalis* venom at a challenge dose of 3× LD50 (20.58 μg/mouse; Figure 7, Appendix A). Although this product offers decent cross-neutralisation against the venoms of several snake species that are not a part of the immunisation mixture, a lack of neutralisation against the venoms of certain species highlights the need to include their venoms in the immunisation mixture.

## 4. Materials and Methods

### 4.1. Sample Details

Snake venoms listed in Table 1 were procured from SA Venom Suppliers (Louis Trichardt, South Africa) from South Africa. Additionally, PANAF-Premium antivenom (batch no. PANAF-016), manufactured against 14 African snake species, namely *B. arietans*, *B. gabonica*, *B. nasicornis*, *B. rhinoceros*, *E. leucogaster*, *E. ocellatus*, *E. carinatus*, *N. haje*, *N. melanoleuca*, *N. nigricollis*, *D. polylepis*, *D. viridis*, *D. jamesoni* and *D. angusticeps*, was procured from Premium Serums and Vaccines Pvt. Ltd. (Pune, Maharashtra, India).

### 4.2. Ethical Statements

Approvals for evaluating the toxicity profiles of venoms and the neutralisation potentials of the antivenom in a mouse model of snake envenoming were granted by the Institutional Animal Ethics Committee (IAEC), Indian Institute of Science (IISc), Bangalore (CAF/Ethics/860/2021; date of approval: 23 June 2022). Experimental protocols were in accordance with the guidelines issued by the Committee for Control and Supervision of Experiments on Animals (CCSEA), Government of India. Male CD-1 mice acquired from Hylasco Biotechnology India Pvt. Ltd. (Hyderabad, Telangana, India) were housed at the Central Animal Facility in IISc. They underwent quarantine for 7 days, following which 3- to 4-week-old mice weighing 18–20 g each were sorted into cages at random (n = 5). The cages were maintained at 18 to 24 °C, with 60 to 65% relative humidity and a 12:12 day/night cycle. WHO’s approved protocols were followed to conduct the preclinical assessment of venoms and antivenoms [1].

### 4.3. Protein Estimation 

The protein content of crude venoms was determined using a modified Bradford method [21] in triplicate (Table 1). Briefly, a working solution of the crude venom (3 µL), reconstituted in molecular-grade water, was incubated with 250 µL of the Bradford reagent (Thermo Fisher Scientific: Waltham, MA, USA). Next, absorbance at 595 nm was measured against a standard bovine serum albumin (BSA) control in an EPOCH2 microplate reader (BioTeK: Santa Clara, CA, USA).

### 4.4. Sodium Dodecyl Sulphate–Polyacrylamide Gel Electrophoresis (SDS-PAGE)

Venoms of African elapid (10 µg) and viperid (12 µg) snakes were subjected to reducing SDS-PAGE in a Mini Gel tank from Bio-Rad (Mini-PROTEAN Tetra Vertical Electrophoresis cell). The gels were stained with Coomassie Brilliant Blue R250 (Sisco Research Laboratories Pvt. Ltd., (Mumbai, Maharashtra, India) and visualised in an iBright CL1000 gel documentation system (Thermo Scientific: Waltham, MA, USA). A prestained marker from Bio-Rad (Precision Plus Protein™ Dual Color standards: Hercules, CA, USA) was used as a reference to determine the approximate molecular mass of proteins [22].

### 4.5. Indirect Enzyme-Linked Immunosorbent Assay (ELISA)

The un vitro binding efficacy of PANAF-Premium antivenom was assessed using indirect ELISA experiments, as described previously [23,24]. Briefly, 96-well ELISA plates were coated with 100 ng of the venom diluted in carbonate buffer (pH 9.6) and incubated overnight at 4 °C. The next day, the plates were washed six times with Tris-buffered saline (0.01 M Tris pH 8.5, 0.15 M NaCl) containing 1% Tween 20 (TBST) to remove unbound venom, followed by the addition of blocking buffer (5% skimmed milk in TBST) and incubation for 3 h at room temperature (RT). Serially diluted PANAF-Premium antivenom was added at the end of incubation with an intermediate round of TBST washing. The plates were incubated overnight at 4 °C, followed by six washes to remove unbound traces of primary antibodies. Further, the plates were incubated for 2 h at RT with horseradish peroxidase (HRP)-conjugated rabbit anti-horse secondary antibody (Sigma-Aldrich, St. Louis, MO, USA), diluted at a ratio of 1:1000 in PBS. The substrate solution (Sigma-Aldrich, USA) containing 2,2-azino-bis (3-ethylbenzthiazoline-6-sulphonic acid) was then added (100 µL/well), and absorbance was measured for 40 min at 405 nm using an Epoch 2 microplate spectrophotometer (BioTeK: Santa Clara, CA, USA). A graph of absorbance versus antivenom dilution was plotted to measure the maximum venom-binding efficacy of the antivenom, taking the reading at the 40th minute. Purified antibodies from naive horses (Bio-Rad Laboratories: Hercules, CA, USA) served as a negative control.

### 4.6. Median Lethal Dose (LD_50_)

The median lethal dose (LD_50_) of the African snake venoms, which corresponds to the minimum amount of venom required to kill 50% of the test population, was determined in the mouse model [1,25]. Briefly, the tail vein of mice (n = 5/group) was intravenously injected with 200 μL of venom diluted in normal saline (0.9% NaCl). In addition, a group of mice received normal saline alone and served as a negative control. The number of dead and surviving mice after 24 h was considered to calculate the LD_50_ of the venom using probit statistics [26]. Details of venom doses (µg), the number of test animals, survival patterns and LD_50_ values with corresponding 95% CIs are provided in Appendix A. GraphPad Prism 9 software was used for the visual representation of results.

### 4.7. Median Effective Dose (ED_50_)

The cross-neutralising potential of PANAF-Premium antivenom was evaluated against the lethal effects of snake venom by determining the median effective value (ED_50_). ED_50_ corresponds to the minimum amount of antivenom required to rescue 50% of the test population when injected with a lethal dose of venom [1,25]. For this experiment, a 3× LD_50_ concentration was used as a ‘challenge dose’, and various dilutions of the antivenom were incubated with this dose for 30 min at 37 °C. Initially, only a single mouse per dose group was used in range-finding experiments. After a 24-h observation period, and based on the results of these range-finding experiments, complete ED_50_ assays were carried out with five dose groups and five mice per dose group. Probit analysis was used to calculate ED_50_ from the observed death and survival patterns of mice. Furthermore, neutralisation potencies were calculated based on the ED_50_ values using the following equation [27]. GraphPad Prism 9 software was used for the visual representation of results.
(1)$$\mathrm{Antivenom}\;\mathrm{neutralisation}\;\mathrm{potency}\;(\mathrm{mg}/\mathrm{mL})=\frac{(n-1)\times \mathrm{LD}50\;\mathrm{of}\;\mathrm{venom}\;(\mathrm{mg}/\mathrm{mouse})}{\mathrm{ED}50\;\mathrm{of}\;\mathrm{antivenom}\;(\mathrm{mL})}$$

Here, n represents the number of LD_50_ values used as the challenge dose.

Additionally, the results were also expressed in terms of the number of LD_50_s neutralised per millilitre of the antivenom by dividing the potency results in mg/mL with the LD_50_ value of the respective venom in μg/mouse.

## 5. Conclusions

Although recombinantly produced monoclonal antibodies (mAbs) are lauded as the future of effective snakebite treatment, considerable research, optimisation and clinical trials are still needed before they can be commercialised. Other alternative strategies to mAbs include aptamers, small-molecule inhibitors, medicinal plant extracts and peptides. Exploiting the cross-neutralisation potential of conventional antivenoms, raised against certain species, for treating bites from closely related snakes offers an economical and immediate solution to snakebites. This is particularly true for the developing regions of the world, such as sSA. Therefore, we evaluated the cross-neutralisation potential of PANAF-Premium antivenom against the venoms of the medically relevant snake species from sSA that were not specifically targeted by this product. Our findings revealed that this antivenom is not only capable of neutralising the venoms that are a part of its immunisation mixture (*N. nigricollis* and *D. polylepis*) but also exhibits significant cross-reactivity towards the venoms of other *Naja* species, including *N. senegalensis*, *N. nigricincta*, *N. nivea*, *N. woodi*, *N. anchietae*, *N. mossambica*, *N. katiensis*, *N. ashei*, *N. nubiae* and *N. pallida*. Similarly, in the case of viperid snakes, the neutralisation potency of this antivenom supersedes its marketed claim against *E. ocellatus* and *B. arietans* venoms that are a part of the immunisation mixture, as well as against the venoms of the untargeted snake species: *E. coloratus* and *E. pyramidum*. The venoms of N. *annulifera* and *B. caudalis* were the only two outliers in the study, against which PANAF-Premium antivenom did not confer neutralisation in a preclinical setting. These results highlight the cross-neutralisation potential of PANAF-Premium antivenom against the venoms of various medically important snakes from sSA, making it a promising product to ameliorate the snakebite burden in this region. However, a thorough clinical investigation is required before this antivenom can be advocated for treating envenoming from non-targeted snake species in this region.

## Figures and Tables

**Figure 1 ijms-25-04213-f001:**
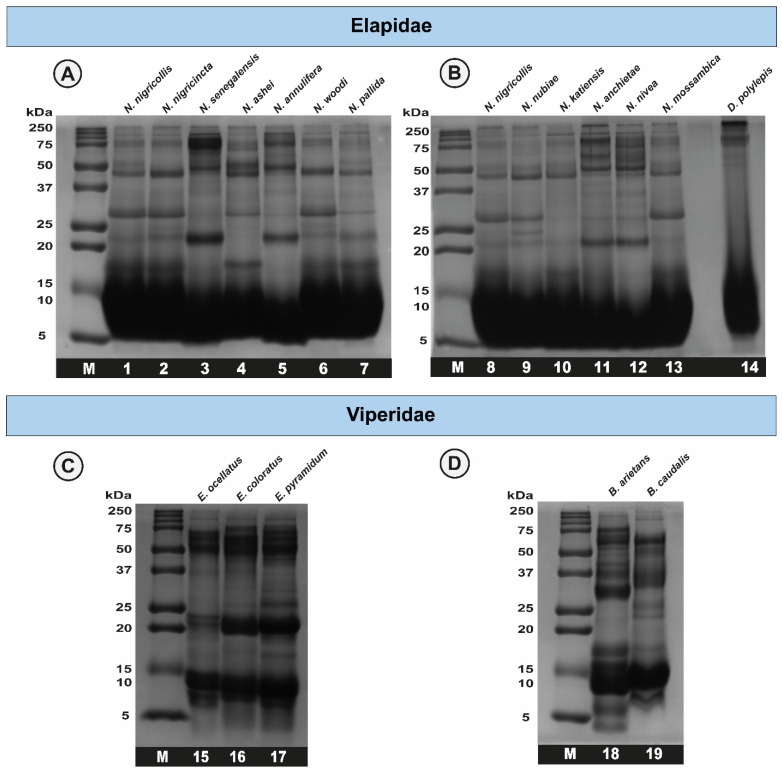
Electrophoretic profiles of snake venoms from sSA. This figure depicts the reducing SDS-PAGE profiles of elapid (**A**,**B**) and viperid (**C**,**D**) snake venoms. **M**: protein marker; **1**: *N. nigricollis*; **2**: *N. nigricincta*; **3**: *N. senegalensis*; **4**: *N. ashei*; **5**: *N. annulifera*; **6**: *N. woodi*; **7**: *N. pallida*; **8**: *N. nigricollis*; **9**: *N. nubiae*; **10**: *N. katiensis*; **11**: *N. anchietae*; **12**: *N. nivea*; **13**: *N. mossambica;* **14**: *D. polylepis*; **15**: *E. ocellatus*; **16**: *E. coloratus;* **17**: *E. pyramidum*; **18**: *B. arietans*; and **19**: *B. caudalis*.

**Figure 2 ijms-25-04213-f002:**
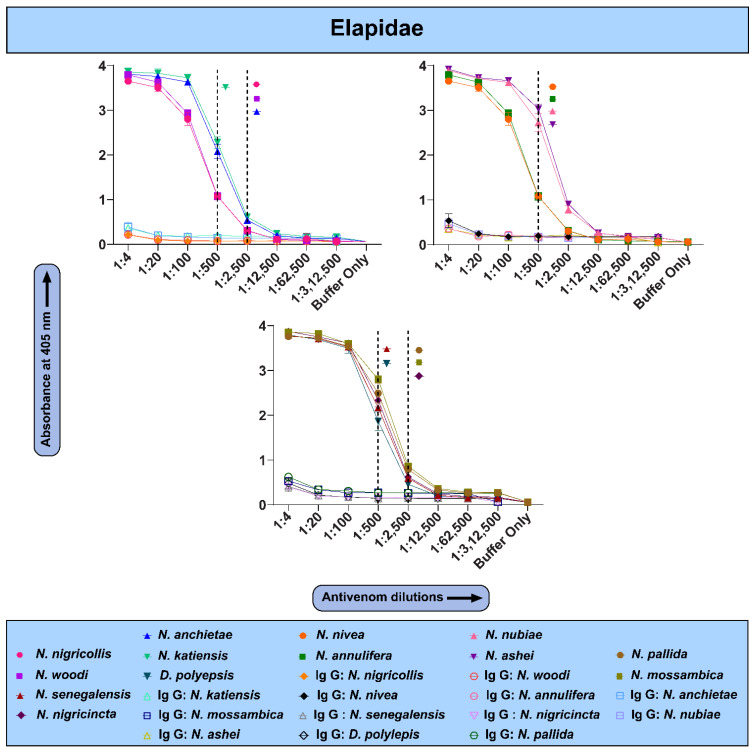
In vitro binding potential of PANAF-Premium antivenom towards sSA elapids. The line graphs represent the binding of naive IgG (1:4) and PANAF-Premium antivenom towards elapid snake venoms from sSA. Absorbance at 405 nm was plotted against various dilutions of antivenom to determine their in vitro venom-binding potential. Each antivenom dilution was tested in triplicate, and the mean of these values was plotted, with the error bars representing standard deviations. Unique colour codes are used to represent individual snake species. The dotted line represents the titre value of the antivenom against the respective venom.

**Figure 3 ijms-25-04213-f003:**
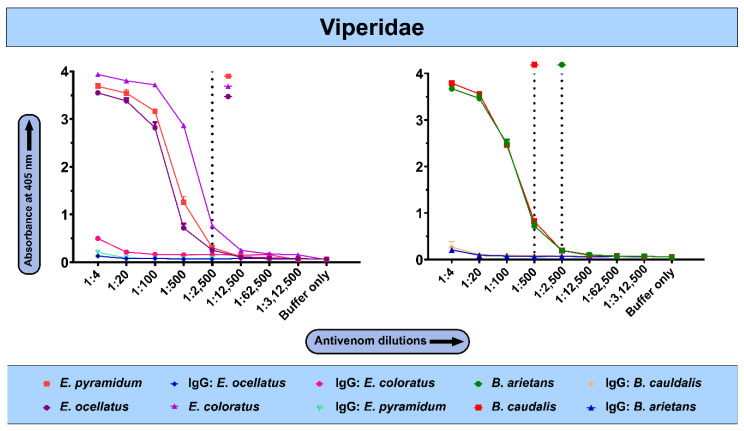
In vitro binding potential of PANAF-Premium antivenom towards sSA viperids. The line graphs represent the binding of naive IgG (1:4) and PANAF-Premium antivenom towards viperid snake venoms from sSA. Absorbance at 405 nm was plotted against various dilutions of antivenom to determine the in vitro cross-reactivity. Each antivenom dilution was tested in triplicate, and the mean of these values was plotted, with the error bars representing standard deviations. Unique colour codes are used to represent individual snake species. The dotted line represents the titre value of the antivenom against the respective venom.

**Figure 4 ijms-25-04213-f004:**
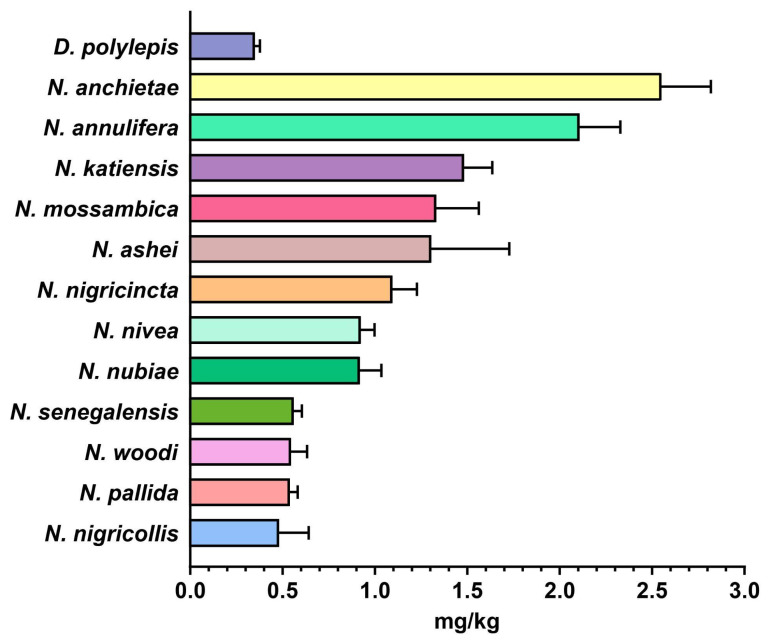
Toxicity profiles of elapid snake venoms. This figure depicts the LD_50_ (mg/kg) of the venoms of various medically important elapid snakes from sSA. The error bars represent the 95% confidence intervals calculated using probit statistics. Details of venom doses (µg), the number of test animals, survival patterns and LD_50_ values with corresponding 95% confidence intervals (CIs) are provided in Appendix A.

**Figure 5 ijms-25-04213-f005:**
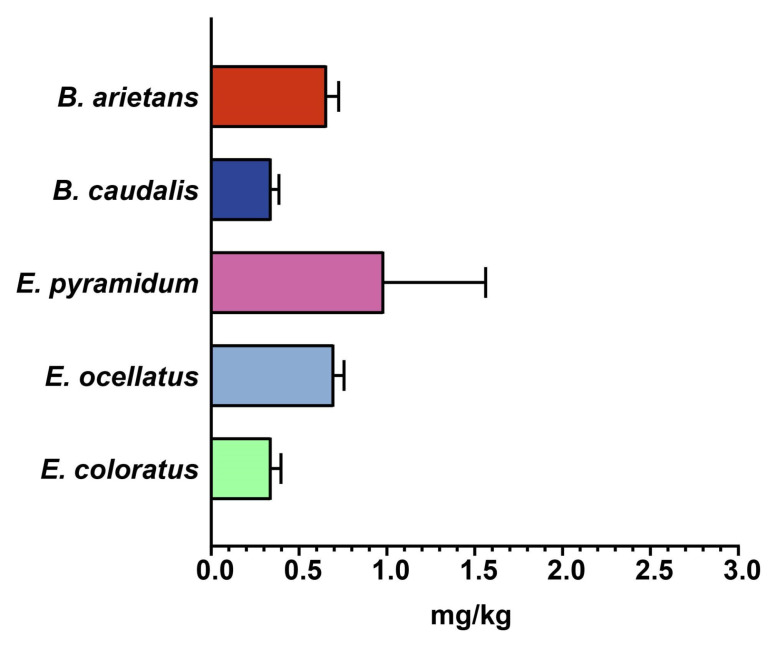
Toxicity profiles of viperid snake venoms. This figure depicts the LD_50_ (mg/kg) of the venoms of various medically important viperid snakes from sSA. The error bars represent the 95% confidence intervals calculated using probit statistics. Details of venom doses (µg), the number of test animals, survival patterns and LD_50_ values with corresponding 95% CIs are provided in Appendix A.

**Figure 6 ijms-25-04213-f006:**
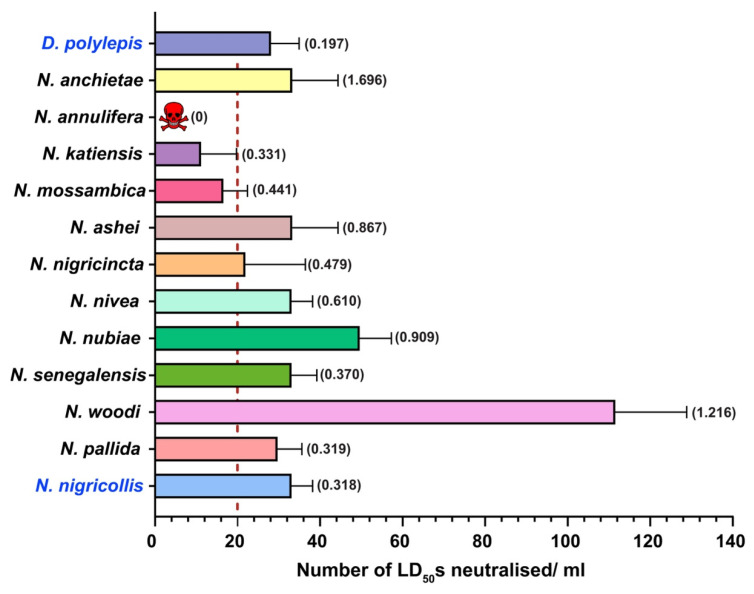
In vivo neutralisation potential of PANAF-Premium antivenom against elapid snake venoms from sSA. The vertical dotted line represents the marketed neutralisation potency of this antivenom against elapid venoms highlighted in blue (≥20 LD_50_s/mL). Additionally, neutralisation potency values in mg/mL against the 3× challenge dose of each venom have also been provided in parentheses. The skull and crossbones symbol indicates complete failure of the antivenom in conferring protection against the *N. annulifera* venom. The error bars representing the 95% CIs were calculated using probit statistics. The details of antivenom doses (µL), survival patterns, ED_50_ values and potency values, along with their corresponding 95% CIs, are provided in Appendix A.

**Figure 7 ijms-25-04213-f007:**
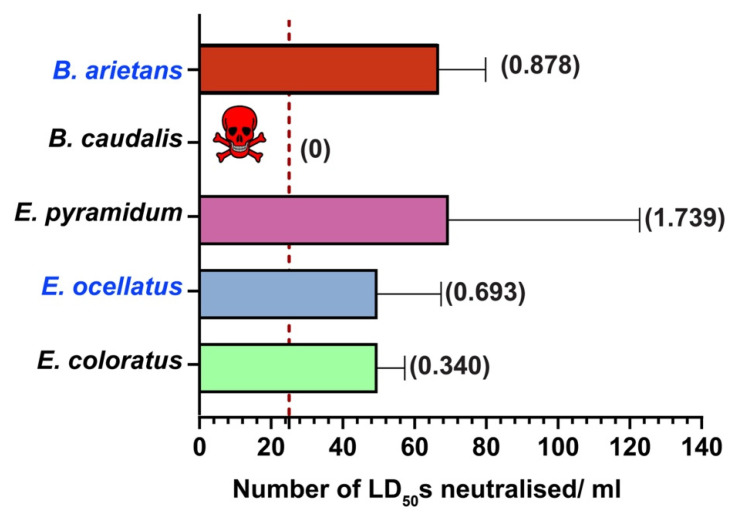
In vivo neutralisation potential of PANAF-Premium antivenom against viperid snake venoms from sSA. The vertical dotted line represents the marketed neutralisation potency of this antivenom against viper venoms highlighted in blue (≥25 LD_50_s/mL). Additionally, neutralisation potency values in mg/mL against the 3× challenge dose of each venom have also been provided in parentheses. The skull and crossbones symbol indicates complete failure of the antivenom in conferring protection against the *N. annulifera* venom. The error bars representing the 95% CIs were calculated using probit statistics. The details of antivenom doses (µL), survival patterns, ED_50_ values and potency values, along with their corresponding 95% CIs, are provided in Appendix A.

**Table 1 ijms-25-04213-t001:** List of venoms analysed in this study. The green cells indicate the species used in the immunisation mixture (species, location and venom concentration), whereas the red cells depict the species against which the cross-neutralisation potential of PANAF-Premium was tested in this study.

Sr. No.	Species	Location	Venom Concentration (mg/mL)
***Naja* spp.**			
1	*N. nigricollis*	Togo, Tanzania, Cameroon	1.39
2	*N. nubiae*	Egypt	1.32
3	*N. katiensis*	Togo	1.26
4	*N. anchietae*	Botswana	1.05
5	*N. nivea*	Western Cape	1.33
6	*N. mossambica*	RSA	1.50
7	*N. nigricincta*	Namibia	1.39
8	*N. senegalensis*	West Africa	1.50
9	*N. ashei*	Northern Cape	1.23
10	*N. annulifera*	RSA	1.49
11	*N. woodi*	Northern Cape	1.42
12	*N. pallida*	Tanzania	1.39
***Bitis* spp.**			
13	*B. arietans*	Zimbabwe	2.43
14	*B. caudalis*	Limpopo	1.41
***Echis* spp.**			
15	*E. ocellatus*	Benin	2.09
16	*E. coloratus*	Egypt	2.58
17	*E. pyramidum*	Egypt	2.06
***Dendroaspis* spp.**			
18	*D. polylepis*	Zimbabwe	0.79

## Data Availability

Data is contained within the article and Appendix A.

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
