# Peer review of "Harnessing the Cross-Neutralisation Potential of Existing Antivenoms for Mitigating the Outcomes of Snakebite in Sub-Saharan Africa"

_ijms, 2024, doi:10.3390/ijms25084213_

Round 1

Reviewer 1 Report

Comments and Suggestions for Authors

Khochare et al. present a prospective study that characterizes the proteomes of medically important venomous snakes (13 elapids, 5 viperids) in Sub-Saharan Africa and then uses a murine model of envenomation and death to assess the efficacy of PANAF-Premium antivenom.

The rationale for the investigation was that manufactures of antivenoms in sub-Saharan Africa have or are discontinuing antivenom production of products effective against vipers in their geographical area. The authors posit that a product made in India that is effective against medically important snakes there may have sufficient cross reactivity with African snakes to substitute for products no longer available there. Thus, the explicit goals of the investigation were to determine the reactivity of PANAF-Premium antivenom in vitro and in vivo as a neutralizing agent for the sSA venoms.

The authors first present protein bands in SDS gels of the various venoms to characterize the similarities and differences amongst the venoms. This is interesting, although not knowing what each band represents is a small issue. The following figures were in general informative and interesting, with in vitro and in vivo efficacy of the antivenom demonstrated. The conclusions reached were consistent with the data presented. I have only a few questions.

Results

Figures 2-3. Please indicate in the figure legend if the value is a mean – the standard deviation is identified. Also, indicate the number of replicates per condition per venom.

Figures 4-6. Indicate in the figure legend if the value is a mean and standard deviation per venom.

Figures 6-7. Explain in the figure legend what the skull and crossbones indicate.

Methods

It is unclear what the composition of the venoms were that were procured as described. Were these venoms that were obtained freshly and then frozen? If frozen, at what temperature were they maintained after harvest, transport, and in the laboratory before experimentation? If lyophilized, what buffer was used to reconstitute them in? My impression is that the venom was unprocessed based on the variable concentrations of protein determined for each venom in table 1. However, most venoms have protein concentrations more concentrated than the samples in table 1. Lastly, if unfiltered, was there particulate in the samples?

What sort of mice were used in terms of strain and weight? While the authors cite the methodology of envenomation of the mice (1,25), was the volume injected the same between venoms? Were the venoms standardized in protein concentration before administration? Also, how did the authors compose their antivenom/venom mixtures to standardize the dose in terms of ml venom mixture/kg mouse?

Please provide the software and vendor of the program used to calculate LD50 values.

Author Response

Khochare et al. present a prospective study that characterizes the proteomes of medically important venomous snakes (13 elapids, 5 viperids) in Sub-Saharan Africa and then uses a murine model of envenomation and death to assess the efficacy of PANAF-Premium antivenom.

The rationale for the investigation was that manufactures of antivenoms in sub-Saharan Africa have or are discontinuing antivenom production of products effective against vipers in their geographical area. The authors posit that a product made in India that is effective against medically important snakes there may have sufficient cross-reactivity with African snakes to substitute for products no longer available there. Thus, the explicit goals of the investigation were to determine the reactivity of PANAF-Premium antivenom in vitro and in vivo as a neutralizing agent for the sSA venoms.

The authors first present protein bands in SDS gels of the various venoms to characterize the similarities and differences amongst the venoms. This is interesting, although not knowing what each band represents is a small issue.

We agree with the reviewer and understand where they are coming from. However, performing mass spectrometry of individual bands of 18 venom samples is beyond the scope of this study.

The following figures were in general informative and interesting, with in vitro and in vivo efficacy of the antivenom demonstrated. The conclusions reached were consistent with the data presented. I have only a few questions.

We thank the reviewer for the kind words.

Results

Figures 2-3. Please indicate in the figure legend if the value is a mean – the standard deviation is identified. Also, indicate the number of replicates per condition per venom.

Following the reviewer’s suggestion, we have now added this information to the figure legend. 

Figures 4-6. Indicate in the figure legend if the value is a mean and standard deviation per venom. 

The median values for toxicity and neutralisation experiments (LD50 and ED50) have been shown with error bars representing the 95% CI calculated using Probit statistics for Figures 4-7. This has now been mentioned in the legends of respective figures.

Figures 6-7. Explain in the figure legend what the skull and crossbones indicate.

In Figures 6 and 7, this symbol indicates a complete failure of PANAF-Premium antivenom in protecting mice injected with N. annulifera and B. caudalis venoms. This has now been mentioned in the figure legends.

Methods

It is unclear what the composition of the venoms were that were procured as described. Were these venoms that were obtained freshly and then frozen? If frozen, at what temperature were they maintained after harvest, transport, and in the laboratory before experimentation? 

Venoms were extracted and immediately snap frozen and stored at -80 0C until lyophilisation.

If lyophilized, what buffer was used to reconstitute them in? 

We have used molecular-grade water for reconstitution. This information has now been added to the manuscript (line  325).

My impression is that the venom was unprocessed based on the variable concentrations of protein determined for each venom in table 1. However, most venoms have protein concentrations more concentrated than the samples in Table 1. Lastly, if unfiltered, was there particulate in the samples?

Venom concentrations often change from sample to sample. Venom samples were not filtered, but no particulate matter was observed.

What sort of mice were used in terms of strain and weight? 

Following the reviewer’s suggestion, we have now incorporated all the requested details regarding the animals utilised in our study in section 5.2, lines 315-321.

While the authors cite the methodology of envenomation of the mice (1,25), was the volume injected the same between venoms? 

Various doses (based on the dry weight) of the venoms were prepared in 0.9% NaCl, and each animal received 200 μL venom dose intravenously, which now has been clarified in line number 361.

Were the venoms standardized in protein concentration before administration?

Various doses of venoms were prepared in saline based on their dry weight. This is a standard practice and follows the WHO preclinical assay.

Also, how did the authors compose their antivenom/venom mixtures to standardize the dose in terms of ml venom mixture/kg mouse?

Following LD50 estimation, the challenge dose of venom equivalent to 3x LD50 was mixed with various dilutions of the antivenom and incubated with these doses for 30 minutes at 37 ℃. This mixture was injected into a single mouse per dilution in a range-finding study. Based on these results, the complete experiment was performed with five mice per dose group. The probit method was used to plot survival patterns to calculate ED50 values. Based on the following formula, the neutralisation potency of antivenom was calculated against each venom.

Antivenom neutralisation potency (mg/ml) = (n-1) LD50 of venom (mg/mouse)ED50 of antivenom (ml)

Here, n represents the number of LD50 used as the challenge dose.

This is now clearly mentioned in the methods.

Please provide the software and vendor of the program used to calculate LD50 values.

We have used the probit method to calculate LD50 values (Finney, D., Probit analysis 3rd; 1971). The GraphPad Prism 9 software was used for the visual representation of results.

Reviewer 2 Report

Comments and Suggestions for Authors

In this study, the authors evaluated the cross-neutralisation potential of Premium Serums’ PANAF-Premium antivenom against medically relevant snakes across sub-Saharan Africa (sSA). The authors concluded that their results highlighted the potential use of PANAF-Premium antivenom in treating bites from diverse snakes across sSA and the utility of harnessing the cross-neutralisation potential of antivenoms.

Comments

This is an interesting study. The reviewer has some concerns as follows:

1.     In the Methods section, some information for the in vivo studies needs to be strengthened, such as source of mice, mouse gender, mouse age, and selected dosage range. Moreover, why the 24 hours were selected to be time course to calculate the LD50 of the venom in mice?

2.     In the in vivo studies, why do LD50 values ​​have standard error values (SD or SE?)? How are these data calculated? It needs to be described in the Methods in detail.

3.     In Table 1, how to decide the venom concentrations? The references can be provided.

4.     In Figures 2 and 3, please explain what is the unit for values in y axis? What do the numbers 1-4 on the y axis of the figure mean?

Author Response

In this study, the authors evaluated the cross-neutralisation potential of Premium Serums’ PANAF-Premium antivenom against medically relevant snakes across sub-Saharan Africa (sSA). The authors concluded that their results highlighted the potential use of PANAF-Premium antivenom in treating bites from diverse snakes across sSA and the utility of harnessing the cross-neutralisation potential of antivenoms.

Comments

This is an interesting study. The reviewer has some concerns as follows:

  1. In the Methods section, some information for the in vivo studies needs to be strengthened, such as source of mice, mouse gender, mouse age, and selected dosage Following the reviewer’s suggestion, we have now incorporated all the requested details regarding the animals utilised in our study in section 5.2 (lines 315-321). The information regarding selected doses for these experiments was already mentioned in the supplementary tables.

Moreover, why the 24 hours were selected to be time course to calculate the LD50 of the venom in mice?

An observation period of 24 hours is a standard for the WHO-recommended in vivo LD50 and ED50 assays.

  1. (WHO), W.H.O. Snakebite envenoming. 2019  [cited 2019; Available from: https://www.who.int/news-room/fact-sheets/detail/snakebite-envenoming.
  2. Saganuwan, S.A.; The new algorithm for calculation of median lethal dose (LD(50)) and effective dose fifty (ED(50)) of Micrarus fulvius venom and anti-venom in mice. Int J Vet Sci Med, 2016. 4(1): p. 1-4.
  3. Gutiérrez et al., 2021; In Vitro Tests for Assessing the Neutralizing Ability of Snake Antivenoms: Toward the 3Rs Principles; Int J Vet Sci Me. 2016 Nov 29;4(1):1-4. doi: 10.1016/j.ijvsm.2016.09.001.

  1. In the in vivo studies, why do LD50 values ​​have standard error values (SD or SE?)? How are these data calculated? It needs to be described in the Methods in detail.

The graphs for these LD50 values represent the median values of these toxicity assays, with error bars denoting the 95% CI, calculated using Probit statistics, as already mentioned in the method section (line 362).

  1. In Table 1, how to decide the venom concentrations? The references can be provided.

This has already been described in section 5.3, along with the reference to Broadford’s assay protocol used in this study. 

  1. In Figures 2 and 3, please explain what is the unit for values in y axis? What do the numbers 1-4 on the y axis of the figure mean?

The y-axis in these ELISA figures represents absorbance values recorded at 405 nm as indicated in the figure.

Round 2

Reviewer 1 Report

Comments and Suggestions for Authors

Thank you for your detailed answers. No further comments.

Reviewer 2 Report

Comments and Suggestions for Authors

This revised manuscript has a great improvement and can be accepted.